# Health and economic burden of canine rabies in Chilga district, centeral Gondar Zone, Amhara Region, Ethiopia

Tewodros Birhan Engida[1,2], Amsalu Misgie Molla [3]*, Wassie Molla[2], Wudu Temesgen Jemberu[2,4]

1 Chilga Animal Resources development and Promotion Office, Chilga Woreda, Ethiopia, 2 Department of Veterinary Epidemiology and Public Health, College of Veterinary Medicine and Animals Sciences, University of Gondar, Gondar, Ethiopia, 3 Department of Veterinary Science, College of Agriculture and Environmental Sciences, Debre Tabor University, Debre Tabor, Ethiopia, 4 International Livestock Research Institute, Addis Ababa, Ethiopia

* amsaluvet21@gmail.com

## Abstract

Rabies is a deadly neglected tropical disease caused by a Lyssavirus that affects all warm-blooded animals, including humans. While canine rabies is prevalent in Ethiopia, its public health and economic burden is not well understood due to a lack of reliable surveillance data. This study aimed to estimate the economic and public health burden of canine rabies in Chilga district, Northwest Ethiopia. The study was conducted retrospectively over two years, from September 2017 to September 2019. A multistage cluster sampling approach was used to select representative households for data collection. A total of 768 households' heads were interviewed to collect data about rabies in their family members, dogs and livestock. The public health burden of the disease was estimated in disability adjusted life years (DALYs), while the economic burden was estimated by aggregating the costs of rabies in humans and livestock using cost estimation equations. The annual human rabies death in the district was estimated at 37deaths/100,000 populations (95%CI:10–95). The overall public health burden of canine rabies in the district was 1576.5 DALYs/100,000 population (95% CI: 424 – 4025) per year. The economic burden of rabies in the district was estimated at 887.3 ETB/household (95%CI: 688.0-1109.2)/two year. The major portion (76.7%) of the economic burden was contributed by livestock losses. Generally, the study revealed a significant public health and economic burden estimates of canine rabies in the district that needs a control intervention focus on dog vaccination, which could have significant impact in reducing both public health and economic burden.

**Data availability statement:** All relevant data supporting the findings of this study are provided within the manuscript and its Supporting Information files.

**Funding:** The author(s) received no specific funding for this work.

**Competing interests:** The authors have declared that no competing interests exist.

## Author summary

Rabies, caused by the Lyssavirus, affects all warm-blooded animals, including humans, and poses a significant public health threat in Ethiopia, particularly due to canine rabies. This study assessed the economic and health burdens of canine rabies using a multistage cluster sampling method to interview 768 household heads. The findings revealed a public health burden of 37 deaths per 100,000 population annually, resulting in 1,576.5 Disability Adjusted Life Years (DALYs) lost per 100,000. The economic burden was estimated at 887.3 Ethiopian birr per household/two year, primarily due to livestock losses, which accounted for 76.7%. The study highlights the urgent need for interventions, particularly dog vaccination, to mitigate the public health and economic burdens of rabies.

## 1. Introduction

Rabies is a neglected tropical disease caused by members of the *Lyssavirus* genus in the *Rhabdoviridae* family [1,2]. It is the most serious zoonotic disease that someone might encounter [3]. Naturally, the virus circulates among animals such as dogs, cats, ferrets, fox, raccoon, mongooses, bats, and wolves [4,5], but it can be spilled over to humans through these animals, particularly those of dogs [6]. The animal and human usually contract rabies from the bite of an infected animal. The virus may enter into the body if the mucous membranes or a scratch or breaks in the skin have contact with saliva containing the rabies virus [7]. The most frequent route by which the virus transmit to human is through an animal bite [6]. A human infection causes fatal encephalomyelitis [8], which is expressed by confusion, aggressive behavior, and extremely painful laryngeal spasms [6,9]. Once clinical signs appear, fatality rate is almost 100% [10].

Estimations on the global burden of rabies indicate a health impact of 59,000 human deaths per year, a loss of 3.7 million DALYs per year and economic loss of about 8.6 billion USD per year mainly due to premature death and post-exposure treatment costs [11]. Most of the human death and economic loss occur in Asian and African countries [12,13]. Rabies also imposes significant economic losses, including estimated livestock losses exceeding 50 million USD annually, given its impact on cattle and other animals [14].

The Rabies burden in Ethiopia was estimated at 3,000 human deaths, 194,000 Disability-Adjusted Life Years (DALYs) and 2 million USD post-exposure treatment costs annually [15,16].

Canine rabies is an economically unique zoonosis because most of its associated costs do not result from illness, but are the consequence of human deaths and efforts to prevent the disease in humans, livestock and companion animals. This unique pattern of costs reflects two basic facts: the case fatality rate of rabies is nearly 100%, and the disease is completely preventable through timely post exposure prophylaxis (PEP) with rabies vaccine [17]. As a result, many individuals who are at very low risk

of developing the disease still seek post exposure vaccination, regardless of the recommendation of health professionals [18].

At present, the burden of the disease is high to people who least afford vaccination. Improving the availability of PEP could reduce the number of human deaths, but it is costly. The incidence of dog-mediated rabies can be reduced by sustained mass dog vaccination, and the cost of PEP will would decrease with time if the vaccination is done appropriately [19]. National dog vaccination programs and better access to PEP require consistent and sustained commitment but will have widespread health benefits, particularly for the poorest communities in the world [19].

Ethiopia is one of the worst rabies affected country [11], with domestic dogs being the major sources of infection to humans. The dog population in Ethiopia is primarily estimated from regional veterinary service records, which indicate that the country has approximately 7.5 million dogs [20,21]. Dog management is poor and anti-rabies dog vaccination is non-existing in most rural areas of Ethiopia, and national estimates show that only about 3% of the dog population in Ethiopia was vaccinated against rabies in 2012, which is far below the 70% coverage recommended by WHO to interrupt [16,20]. The large dog population size in combination with poor dog management contributes to a high endemicity of canine rabies in Ethiopia [22]. In Ethiopia, the official annual reports from 2010–2012 rabies diagnosis and surveillance estimate, indicates 12 exposure cases and 1.6 rabies deaths per 100,000 populations [20]. However, the actual numbers are expected to be higher as many cases are not reported [21].

In Ethiopia, rabies remains a neglected zoonotic disease with significant gaps in surveillance. While animal bite exposures are recorded at health facilities for post-exposure prophylaxis (PEP), rabies itself is not consistently a fully notifiable disease, and most cases are diagnosed clinically rather than confirmed due to limited diagnostic capacity and centralized laboratory services. Rabies deaths are under-reported because many victims, particularly in rural areas, do not seek formal care, rely on traditional treatments, or die outside the health system. Weak coordination between public health and veterinary sectors, limited community awareness, and competing health priorities further contribute to incomplete documentation [20]. Dog populations are estimated indirectly using household surveys, dog-to-human ratios, and localized studies, complicated by the large number of free-roaming and ownerless dogs. These factors collectively lead to underestimation of rabies incidence and mortality and explain regional variation in reported data. The widespread traditional practices of handling rabies cases might interfere with medical treatment, resulting in an underreporting of the actual number of rabies cases and its related health burden [23].

This study was aimed at estimating the public health and economic burden of canine rabies in Chiliga district, central Gondar Zone, northwest Ethiopia.

## 2. Materials and methods

### 2.1. Ethics statement

This study was conducted with ethical approval from the University of Gondar, College of Veterinary Medicine and Animal Sciences Research Ethics Review Committee (Ref. No. 005/2017). Informed oral consent was obtained after respondents were informed about the study purpose, interview procedures, and use of their responses, and participation was entirely voluntary with the right to withdraw at any time. Permission to conduct the interviews was obtained from the district health and livestock development and promotion offices.

### 2.2. Study area

The study was conducted in Chilga district, central Gondar administrative zone of Amhara region in Ethiopia (Fig 1). The altitude of the study area ranges between 900–2250 m.a.s.l. The district is characterized by mean annual temperature which ranges between 12°C and 29°C and the mean annual rainfall varies from 950 ml to around 1100 ml. Based on the 2007 national census conducted by the Central Statistical Agency of Ethiopia [24], the district has a total of 32,496 households with a population of 237,581(120,103 male and female 117,478 female). Livestock play an important role for

## Map of study area

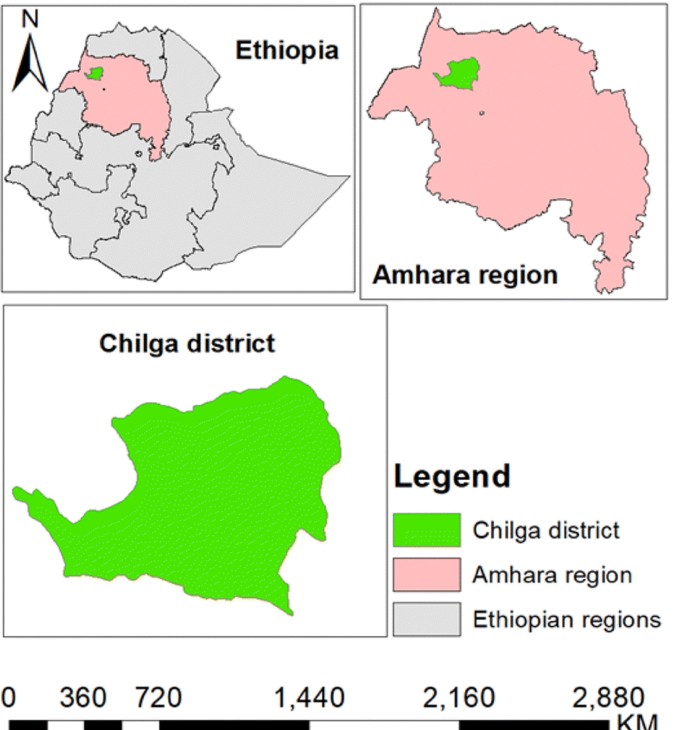

**Fig 1. Map of the study district.** The map was created by authors using the base map shapefiles for Ethiopian subnational administrative boundaries available in OCHA-FISS [27].

income-generating activities of the community in the rural area. In 2019, the total animal population in the district was estimated at 1,195,136 (524,230 cattle, 20,882 equines and 650,024 small ruminants) [25]. There are 49 rural and 6 town kebeles in the district [26].

### 2.3. Study design

The study was retrospective study in which rabies related data for the preceding two years (September 2017 to September 2019) were collected through questionnaire survey. Samples of household heads in the study district were interviewed about their experience on the occurrence of rabies and associated deaths in their family and livestock to estimate the public health and economic burden of the disease.

**2.3.1. Sampling techniques and sample size determination.** Multistage cluster sampling technique was used to select representative households for data collection. First kebeles from the study district were selected by simple random sampling techniques and then from selected kebeles households were selected systematically. For the selection of kebeles, first the list of 55 kebeles present in the district was obtained from the district office, and about half of them (23 rural and 2 town), which are believed to be enough to represent the district, were selected randomly using the MS Excel rand function. Then the total number of households in the selected 25 kebeles was obtained from the district agriculture office, which contained 7,680 households. The sampling interval was then calculated as 7,680 divided by the sample size 768, resulting in a sampling interval of 10. In each kebele, the first household (1st) was selected from the road entrance into the kebele, and sampling continued for the remaining households along the road.

The sample size was determined using Cochran's sample size formula [28].

$$n = \frac{t^2 * pq}{d^2} \cdots \cdots , \quad n = \frac{1.96^2 * 0.5 * (1 - 0.5)}{0.05^2} = 384$$

Where,'t' is the value for selected alpha level of 0.025 in each tail = 1.96. (p), (q) is the estimate of variance = 0.25 (assuming 50% of households would experience the disease), d is the acceptable margin of error = 0.05.

The above sample size works for simple random sampling techniques. Nevertheless, the sampling technique used in the study was multistage cluster sampling and the sample size was adjusted for clustering effect by doubling of the calculated sample size, i.e., 768, rule of thumb used when detail information about intercluster variance is absent [29].

**2.3.2. The questionnaire survey.** The sampled households in the kebele were interviewed using a structured questionnaire (S1 File) about their experience of livestock and human rabies in the preceding two years. The questionnaire was developed in English and was translated to the local language, Amharic. The first question of the questionnaire was posed to ascertain the familiarity of respondent with rabies. That was done by asking the respondent whether he/she is familiar with rabies by using the local name ('Ye-ebid wusha beshita') and the main clinical and epidemiological feature of the disease and the response was compared with literature description of rabies clinical signs such as aggressiveness, restlessness, salivation and paralysis. If the answer matched the clinical signs description, the respondent would continue with other part of the questionnaire and if not was excluded from the interview. The remaining part of the questionnaire covered experience of the respondents about rabies exposure, death and treatment/preventive measures (both modern and traditional) in their family members and livestock holding. The respondents reported rabies exposures were categorized into two major categories. The first is less risky exposure which included respondents claimed exposures which could not pose risk of disease. This roughly include WHO exposure categories of "Category 0" (perceived exposure without physical contact), and "Category I" (touching or feeding animals or licks on intact skin). The second category is risky exposure that roughly include WHO exposure "Category II" (nibbling of uncovered skin, minor scratches or abrasions without bleeding) and "Category III" (single or multiple transdermal bites or scratches, contamination of mucous membranes or broken skin with saliva) [30]. The questionnaire also included questions related to cost of treatment and livestock deaths or culling (S1 File).

## 2.4. Estimating the public health burden of Canine Rabies

Disability-adjusted life years (DALYs) is a method developed by the World Health Organization [31], to calculate the rabies health burden. Disability adjusted life years is the sum of years of life lost (YLL) due to premature death and years lived with disability (YLD) associated with the disease.

YLL is calculated as:-

$$YLL = \sum (NDeath_i * LE_i)$$

Where; Ndeath_i is number of human deaths within age category (i) and LE_i is life expectancy of the concerned age category (i). Eighteen age groups with a 5 years age interval from 0 to 85 years and above were Considered (i = 1... 18) according to the age classification of WHO life expectancy table [32].

YLD is calculated as:-

$$YLD = (NDis_i * D * DW)$$

Where, Ndis_i is total number of disabilities in the the concerned age group (i), with the D is duration of the disability in years and DW is Weight. In this study the disability considered was adverse effect associated with the use of nervous

tissue based rabies vaccine which has disability weight of 0.613 [12]. But no adverse effect was reported in the study and hence YLD was zero.

## 2.5. Estimating the economic burden of Canine Rabies

The economic burden of the disease associated with both humans and livestock was estimated using the following cost models by adapting from Shwiff *et al.* [33] and Rushton [34].

*Cost of human rabies:*

The total cost of human rabies (TCRH) was estimated using Equation (Eq.1).

$$TCRH \ = \ NEm * (DMC + \ INMC) \ + NEt \ (CDPH + CTPH + ACPH) \tag{Eq.1}$$

Where; NEm = Number of exposed humans that attended modern treatment, DMC = Direct medical cost of rabies exposed human, INMC = Indirect non-medical cost of rabies exposed human, NEt = Number of exposed human that attended traditional treatment, CDPH = Cost of diagnosis per human in traditional treatment, CTPH = Cost of traditional treatment per human, ACPH = Additional costs per human when come to traditional healers (costs of transportation, food).

*Cost of rabies in livestock:*

The study was considered four livestock species (cattle, equine, goat and sheep), but the disease was reported only in cattle and equines. The total value loss of livestock due to rabies (TCRL) or each livestock species was estimated using Equation (Eq.2).

$$TCRL = \sum in = 1[Price1 - \ Price2] \tag{Eq.2}$$

Where; n = number of animals exposed, potential price 1 = price before exposure, Price2 = culled/salvage value. The cost of livestock price was the sum of cost of rabies in cattle and cost of rabies in equines in Livestock price per head. Prices of animals were based on respondents' estimate. Price 2 for equines is obviously zero as rabid equines have no salvage value.

*The total cost (TC) associated with rabies*

The total cost (TC) associated with rabies was estimated by considering the sum of human, and livestock components using Equation (Eq.3).

$$TC \ = \ TCRH \ + \ TCRL \tag{Eq.3}$$

## 2.6. Data management and analysis

After data collection the data was entered into Microsoft Excel. Public health burden was analyzed in terms of incidences of exposure cases and DALYs per 100,000 population per year. The exposure cases and DALY reported by the study respondents was expressed per 100,000 population based on the ratio of cases or DALYs per the total number of people in the sample, i.e., 768 households with average household size of 7 which is equal to 5376 people. A 95% confidence interval (CI) for the incidence of exposures cases was estimated using exact method of poison probability as [35]:

$$\left( 95\% \ \text{CI} \ = \ \frac{\chi^2_{0.025, 2X}}{2T}, \frac{\chi^2_{0.975, 2X+2}}{2T} \right)$$

where $\chi^2_{p,df}$ is the p-th quantile of the chi-square distribution with df degrees of freedom, X number of cases (exposure cases, deaths), and T person years of duration of the incidence. The exact methods was used because of small number of cases specially death cases.

The 95%CI for DALYs was computed by multiplying the number of cases leading to DALY (deaths in this study case) in the 95% CI calculated above by the average number of DALYs per case.

Economic cost estimates from various sources (human PEP cost, traditional treatment cost, and livestock losses) were averaged over the individuals and sample households. However, the distribution of these costs were not normally distributed as most of the households did not experience rabies exposure and had zero costs. Therefore, bootstrapping method [36] was used to construct confidence interval over the sample households to extrapolate the cost findings to the district population of households.

The economic costs were estimated in Ethiopian Birr (ETB) of the 2017/8–2018/9 value. The average exchange rate of USD to ETB during the study period was 1USD = 28 ETB (www.theglobaleconomy.com/Ethiopia/Dollar_exchange_rate/).

## 3. Results

### 3.1. Public health burden of Rabies

During the study, a total of 768 head of households (respondents) were interviewed about occurrence of rabies in their family and livestock, and anti-rabies traditional practice in Chilga district, Central Gondar administrative zone. Before the main questions, the respondents were first checked about their familiarity about rabies using an operation definition of rabies as set as the question in the questionnaire and all of the responders were found to be familiar for the disease. From the total number of respondents, 222 (28.9%) households reported suspected rabies exposure in their family member in the two years study period. In these households, 424 suspected rabies exposure cases were reported during the 2 years study period, of which 375 were less risky exposure and 49 were risky exposure and 49 were risky exposure. Among 768 respondents having average household size of 7 people, the disease results 456 risky exposures/100, 000 population per year (95%CI: 337–602) (Table 1). Out of the 49 risky exposed cases, 43 (88%) victims visited health centers and 5(10%) visited traditional healers to get traditional diagnosis and treatment for rabies. From those who visited health centers, 32 people received complete dose of PEP, 9 incomplete dose and 2 did not take the PEP. A total of 4 rabies deaths were reported by respondents, resulting in 37 deaths/100, 000 population per year in the district.

In the 49 risky exposures, dog bite sites include: 1 in the head/neck region, 10 on arms/hands, 33 on the legs and 5 on the trunk (Table 2). From the total four deaths, three died following bite of legs, whereas one death was due to hand bite. Three of the dead persons did not receive PEP, while the remaining one started PEP but died before completing the dose. The age of died persons were between 15 and 44 years.

**3.1.1. Health burden estimation in terms of disability adjusted life years (DALYs).** The total DALYs due to YLL was estimated at 1576.5 DALYs/100,000 population per year (95% CI: 424 – 4025) and the highest YLL was found in the age group between 15–19 in male, whereas between 35–39 in female (Table 3).

Health burden is expressed in terms of standard disability adjusted life years (DALYs) and normally calculated as in the summation of YLL and YLD. However, in our analysis years of life lived with disability (YLD), which was considered from Nervous Tissue Vaccine (NTV) adverse effect, was zero due to the absence of NTV adverse effects report in the district during the study period and the DALY was the same as YLL which was 1576.5 DALYs/100,000 population per year in district. Higher DALYs was estimated in males (56%) than females (44%) (Table 3).

**Table 1. The nature of rabies exposure and deaths from 768 households of Chilga district.**

| Rabies related events | 2017/18 | 2018/19 | Total cases | Cases/100, 000 population per year (95%CI) |
|---|---|---|---|---|
| Less risky exposure | 142 | 233 | 375 | 3488 (3,143, 3,859) |
| Risky exposure | 22 | 27 | 49 | 456 (337, 602) |
| Deaths | 3 | 1 | 4 | 37 (10, 95) |

**Table 2. Rabies suspected cases (n = 49) bite sites (head, hand, trunk, and leg), age category and death status.**

| Age categories | Bite sites of the victims and death | | | | |
| --- | --- | --- | --- | --- | --- |
| | Head/neck | Hand | Trunk | Leg | Total |
| 0-4 | 0 | 0 | 1 | 1 | 2 |
| 5-14 | 0 | 1 | 2 | 8 | 11 |
| 15-29 | 1 | 6 | 1 | 15* | 23 |
| 30-44 | 0 | 3* | 1 | 5** | 9 |
| 45-59 | 0 | 0 | 0 | 2 | 2 |
| 60-69 | 0 | 0 | 0 | 2 | 2 |
| 70-79 | 0 | 0 | 0 | 0 | 0 |
| 80+ | 0 | 0 | 0 | 0 | 0 |
| Total | 1 | 10 | 5 | 33 | 49 |

*one death.

**two death.

**Table 3. The estimation of years of life lost (YLLs) due to premature death from 2017/8-2018/9 (N = 5376 people (768 households * 7people)).**

| Age categories | Male | | | Female | | | Person |
| --- | --- | --- | --- | --- | --- | --- | --- |
| | No of death | Life expectancy | YLLs | No of death | Life expectancy | YLLs | Total YLLs |
| 0-4 | 0 | 65.7 | – | 0 | 68.8 | – | – |
| 5-9 | 0 | 62.9 | – | 0 | 66 | – | – |
| 10-14 | 0 | 58.7 | – | 0 | 61.7 | – | – |
| 15-19 | 1 | 54.1 | 54.1 | 0 | 57 | – | 54.1 |
| 20-24 | 0 | 49.7 | – | 0 | 52.5 | – | – |
| 25-29 | 0 | 45.5 | – | 0 | 48 | – | – |
| 30-34 | 1 | 41.2 | 41.2 | 0 | 43.5 | – | 41.2 |
| 35-39 | 0 | 37 | – | 1 | 39.2 | 39.2 | 39.2 |
| 40-44 | 0 | 32.9 | – | 1 | 35 | 35 | 35 |
| 45-49 | 0 | 28.9 | – | 0 | 30.9 | – | – |
| 50-54 | 0 | 24.9 | – | 0 | 26.8 | – | – |
| 55-59 | 0 | 21.1 | – | 0 | 22.8 | – | – |
| 60-64 | 0 | 17.4 | – | 0 | 18.8 | – | – |
| 65-69 | 0 | 14 | – | 0 | 15.1 | – | – |
| 70-74 | 0 | 10.9 | – | 0 | 11.8 | – | – |
| 75-79 | 0 | 8.3 | – | 0 | 8.9 | – | – |
| 80-84 | 0 | 6.1 | – | 0 | 6.5 | – | – |
| 85+ | 0 | 4.4 | – | 0 | 4.8 | – | – |
| Total | 2 | | 95.3 | 2 | | 74.2 | 169.5 |

### 3.2. Economic Burden of Rabies

**3.2.1. Estimated economic costs associated with humans rabies.** A total of 41 rabies-exposed individuals from 38 households out of the 768 sampled households received modern (PEP) treatment over the two study years. The average cost of direct medical and indirect medical costs per exposed person, exposed household and any household in the district are given in table (Table 4). The average cost of PEP treatment at household level in the district was estimated 145. 2 ETB (95%CI: 94–202) per two years.

**Table 4. The costs associated with human rabies cases in seeking PEP over the two study years (2017/8 to 2018/9).**

| Subject | Number | Average medical cost (ETB) | Indirect cost (ETB) | Total Cost (ETB) |
|---|---|---|---|---|
| Exposed people | 41 | 1398.4 | 1314.2 | 2740.6 |
| Affected household | 38 | 1508.8 | 1418.0 | 2926.8 |
| Any household in the district | – | 74.9 (95%CI: 45.96 -108.68) | 70.3 (95%CI: 44.6 - 99.2) | 145.2 (95%CI: 94.1-202.5) |

A total 200 households from the sample of 768 households, who feel exposed to rabies, sought traditional treatment that involves diagnosis and treatment. In these households 395 people got diagnosis, 387 got treatment and 323 incurred additional other costs such as transport and food costs during the treatment in the two years study period. The average traditional treatment costs per 'exposed person', 'exposed household' and any household in the district are shown in Table 5. The average cost of traditional treatment per household in the district was estimated at 61.1 birr (95% CI: 55.6 - 74.9) per two years.

**3.2.2. Estimated economic costs associated with livestock rabies.** Cattle and donkey were the most frequently bitten animals by a suspected rabid dog. The estimated direct economic losses due to the occurrence of rabies in livestock were, therefore, associated with the number of exposed and died cattle and equine due to rabies. Exposed animals might or might not develop rabies. However, in case of cattle, all exposed animals were slaughtered and the meat sold to their neighbors as a precautionary measure, as owners did not want to face the risk that their cattle gradually develop clinical signs. But there was no such precautionary slaughter in donkeys. In this study, all exposed donkeys ended up with death. The cost estimated for livestock included the precautionary cull in cattle and death in donkeys. Households with culled cattle and died donkeys and associated losses are indicted in Table 6. The average cost of rabies in livestock per household in the district was estimated at 681.5 (95% CI: 488.4 - 894.9) Ethiopian birr per two years.

**3.2.3. Total annual economic cost due to rabies in the district.** The overall economic cost of canine rabies in the study district from human exposure (PEP and traditional treatment) and livestock exposure (cull and death) was estimated at 887.3 birr/per household (95% CI: 688.0 -1109.2) over the two study years, which is 443.7ETB/household/year. The livestock associated cost constitutes the major share (76.8%) of the total cost at household level (Fig 2).

## 4. Discussion

This study reported a high burden of rabies in which 28.9% of sample households reported a suspected rabies exposure in their family member with total number of 424 estimated rabies exposures and 37 deaths/100,000 population per year. This reported death is higher than 0.7, 4.2 and 2.9 deaths/100,000 population per year reported respectively in Bishoftu, Lemuna bilbilo and Yabelo districts in other parts of Ethiopia [16]. The current study is also reported higher number of deaths than the previous national estimate in Ethiopia [20], which was 1.6 rabies deaths/100,000 population. The current estimate is also high when compared to estimates from other countries such as the 914 human deaths reported from 2005 to 2014 in Vietnam, where the death rate was 0.11rabies deaths/100,000 population [33]. The community-based interview approach used in this study, despite some potential recall bias, likely captured incidents that may have been overlooked

**Table 5. Costs associated with traditional antirabies treatment over the two study years (2017/8 to 2018/9).**

| Source of cost | Number of 'exposed' peoples | Number of 'exposed' households | Average cost per 'exposed' person | Average cost per 'exposed' household | Average cost for any household in the district |
|---|---|---|---|---|---|
| Diagnosis | 395 | 200 | 40.1 | 85.1 | 22.2 (95%CI:18.8 - 25.6) |
| Treatment | 387 | 200 | 68.7 | 130.9 | 34.1(95%CI: 29.0 - 39.4) |
| Other | 323 | 200 | 21.1 | 34.1 | 8.9 (95% CI: 7.3 - 10.6) |
| **Total** | – | – | **130.0** | **250.1** | **61.1(95%CI: 55.6 -74.9)** |

**Table 6. Costs associated with livestock rabies over the two study years (2017/8 to 2018/9).**

| Species of livestock | Number of 'exposed' animals | Number of 'exposed' households | Average cost per 'exposed' animal | Average cost per 'exposed' household |
|---|---|---|---|---|
| Cattle only | 56 | 45 | 7633.0 | 9498.9 |
| Donkey only | 16 | 16 | 2425.0 | 2425.0 |
| Cattle and donkey | 11 | 5 | 5254.5 | 11560 |

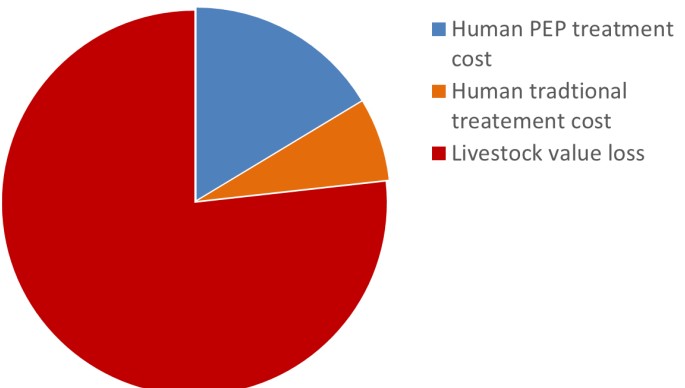

Legend:
- ■ Human PEP treatment cost
- ■ Human tradtional treatement cost
- ■ Livestock value loss

**Fig 2. Composition of human and livestock rabies costs.**

in official reporting systems. This could explain the high number of deaths reported in the current study, but caution is needed as this is a small study that covered small geographical areas and time span.

Among the exposure reports, 88% were less risky exposures, in which the people take unnecessary measures due to lack of good knowledge about the nature of rabies transmission. These were not true exposures, but people were afraid and had taken measures mostly traditional medicines/practices. Creating awareness about risky exposures and non-risky exposures associated with rabies to the community would avoid unnecessary worry and measures during rabies incident.

Almost all DALYs were lost due to premature human deaths. Different reasons for not visiting health facilities and taking PEP on time were mentioned by respondents for these rabies caused deaths. The family for two death victims did not recognize the biting dog was affected by the disease. Another one victim started PEP late and showed clinical sighs while taking the PEP. The other victims had done nothing due to inaccessibility of PEP vaccine.

The total public health burden in the district due to rabies was estimated to be1576.5 DALYs/100,000 population per year. This is high, when compared to 48.7, 297.5 and 239.8 DALYs/100,000 population per year reported by Beyene *et al.* [16] at Bishoftu, Lemuna-bilbilo and Yabelo districts respectively. When DALYs were compared between male and female, DALYs in males (886.4 DALYs/100,000 population per year) was higher than in females (690.1 DALYs/100,000 population per year). This DALY difference is related to the difference in the age of death (male was died at younger age than female) not to the number of death (equal number of people died in each sex category).

This study had also provided an estimate of the direct and indirect economic costs due to rabies in the district during the study period. The total annual economic cost was estimated to be 887.3 birr/per household over the two study years, which is 443.7ETB/household/year. The largest portion (76.8%) of rabies costs in the district was associated with livestock losses. the current estimate of the annual economic cost associated with rabies was substantial, possibly as a result of the district's extremely low dog vaccination coverage. This indicated that there were low rabies control and prevention activities in the study area.

The human rabies cost estimated in this study included only expenditure in the form of medical and non-medical expense. The economic impact of human rabies goes beyond expenses by causing productivity loss due to illness and premature death but this was not captured in this study as it involves complex data and analysis which was outside of the study scope.

This study was done by interviewing a sample of households about their experience of rabies related events in their family member and livestock two years preceding the interview. This might introduce recall bias in the data collected but given the dramatic nature of the disease and obvious symptoms and invariably fatal consequence, the recall problem would be in consequential. This has been similarly justified in the published literatures [37,38]. The operational definition used to check the familiarity of respondents to rabies before proceeding with main data collection from them was narrower than the official rabies case definition by WHO [30], which might cause undue exclusion of respondents. But all responders were found familiar to the disease by the operation definition set, the risk of undue exclusion was absent.

Generally, the study showed a significant public health and economic burden of the disease in the district that needs a control intervention specially strengthening dog vaccination, which was observed to be very low during the study, could have significant impact in reducing both public health and economic burden. Further study based on follow up of a confirmed outbreak of rabies is needed to refine the estimates of public health and economic burden.

## Supporting information

**S1 File. Questionnaire.**
(DOCX)

**S1 Data. Raw Data.**
(XLSX)

## Acknowledgments

The authors would like to thank the households who participated in the study and for providing information needed for the study and the University of Gondar was highly acknowledged for providing the facilities needed to accomplish this study.

## Author contributions

**Conceptualization:** Tewodros Birhan Engida, Wassie Molla, Wudu Temesgen Jemberu.

**Data curation:** Tewodros Birhan Engida.

**Formal analysis:** Tewodros Birhan Engida, Amsalu Misgie Molla, Wudu Temesgen Jemberu.

**Investigation:** Tewodros Birhan Engida, Amsalu Misgie Molla.

**Methodology:** Tewodros Birhan Engida, Amsalu Misgie Molla, Wassie Molla, Wudu Temesgen Jemberu.

**Project administration:** Amsalu Misgie Molla.

**Resources:** Amsalu Misgie Molla.

**Software:** Tewodros Birhan Engida, Amsalu Misgie Molla, Wassie Molla, Wudu Temesgen Jemberu.

**Supervision:** Tewodros Birhan Engida, Amsalu Misgie Molla, Wudu Temesgen Jemberu.

**Validation:** Amsalu Misgie Molla, Wassie Molla, Wudu Temesgen Jemberu.

**Writing – original draft:** Tewodros Birhan Engida, Amsalu Misgie Molla.

**Writing – review & editing:** Tewodros Birhan Engida, Amsalu Misgie Molla, Wassie Molla, Wudu Temesgen Jemberu.

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
