## [Decision Letter · Decision Letter 0]

24 Nov 2025

PNTD-D-25-01493

Health and Economic Burden of Canine Rabies in Chilga District, Centeral Gondar Zone, Amhara Region, Ethiopia

Dear Dr. Molla,

Thank you for submitting your manuscript to PLOS Neglected Tropical Diseases. After careful consideration, we feel that it has merit but does not fully meet PLOS Neglected Tropical Diseases's publication criteria as it currently stands. Therefore, we invite you to submit a revised version of the manuscript that addresses the points raised during the review process.

Please submit your revised manuscript within by Jan 22 2026 11:59PM. If you will need more time than this to complete your revisions, please reply to this message or contact the journal office at plosntds@plos.org. Please include the following items when submitting your revised manuscript:

We look forward to receiving your revised manuscript.

Kind regards,

David Safronetz, Ph.D.

Section Editor

David Safronetz

Section Editor

Shaden Kamhawi

co-Editor-in-Chief

Paul Brindley

co-Editor-in-Chief

**Journal Requirements:**

4) We notice that your supplementary information (Appendix 1) is included in the manuscript file. Please remove them and upload them with the file type 'Supporting Information'. Please ensure that each Supporting Information file has a legend listed in the manuscript after the references list.

Potential Copyright Issues:

i) Figure 1. Please (a) provide a direct link to the base layer of the map (i.e., the country or region border shape) and ensure this is also included in the figure legend; and (b) provide a link to the terms of use / license information for the base layer image or shapefile. We cannot publish proprietary or copyrighted maps (e.g. Google Maps, Mapquest) and the terms of use for your map base layer must be compatible with our CC BY 4.0 license.

6) When completing the data availability statement of the submission form, you indicated that you will make your data available on acceptance. We strongly recommend all authors decide on a data sharing plan before acceptance, as the process can be lengthy and hold up publication timelines. Please note that, though access restrictions are acceptable now, your entire data will need to be made freely accessible if your manuscript is accepted for publication. This policy applies to all data except where public deposition would breach compliance with the protocol approved by your research ethics board. If you are unable to adhere to our open data policy, please kindly revise your statement to explain your reasoning and we will seek the editor's input on an exemption. Please be assured that, once you have provided your new statement, the assessment of your exemption will not hold up the peer review process.

**Comments to the Authors:**

**Please note that one review is uploaded as an attachment.**

**Reviewers' Comments:**

Reviewer's Responses to Questions

**Key Review Criteria Required for Acceptance?**

**Methods**

-Are the objectives of the study clearly articulated with a clear testable hypothesis stated?

-Is the study design appropriate to address the stated objectives?

-Is the population clearly described and appropriate for the hypothesis being tested?

-Is the sample size sufficient to ensure adequate power to address the hypothesis being tested?

-Were correct statistical analysis used to support conclusions?

-Are there concerns about ethical or regulatory requirements being met?

Reviewer #1: The objectives of the study are clearly stated. The study was descriptive, so a hypothesis is not relevant (and was not stated). The design, a survey of households selected using a multistage cluster sample, is appropriate to address the objectives, and the sample size is of sufficient power.

Reviewer #2: The manuscript "Health and Economic Burden of Canine Rabies in Chilga District, Central Gondar Zone, Amhara Region, Ethiopia" (PNTD-D-25-01493) estimates the burden of rabies in DALYs and the economic cost per district. The estimates are higher than previous ones and will be essential for advocating preventive measures against rabies in Ethiopia. However, the paper needs to be strengthened further by incorporating the following elements and by addressing the comments in the attached file.

1. Sampling: The paper needs to better explain the sampling stages. Although the authors doubled the sample size for clustering effect, they have not commented on the homogeneity or heterogeneity of the kebeles. Additionally, they did not estimate the sample using previous estimates and should provide a rationale for this. How the households who refused the survey, were handled needs to be explained. Also, how did they arrive at 25 sample number for kebeles ? Did they use population proportionate to size or something? There is also a need for an explanation of how households within each kebeles were selected? Did they have a list of all households and use random sampling, or follow systematic sampling (every 10th household), or some GIS-based method? Furthermore, were the 768 households evenly divided among the 25 kebeles? The paper would be strengthened if these questions are addressed in the methods section.

2. Extrapolation of sample findings and uncertainty of estimates: The methods section should explain how the findings from the sample were extrapolated to the district or population level. The tables should also describe the extrapolation results. The assumptions and formulas used for extrapolation should be clearly stated in the methods section. Also, there are uncertainties in the findings (95% CI) and the cost (95% CI). These should be stated, and the extrapolated estimates of cost should also reflect these uncertainties and include confidence intervals. The sources of data for costs, such as average medical costs, vaccine costs, etc., and other parameters used should be listed (preferably in a separate table).

3. Inclusion of Dog Rabies Vaccination Costs: Vaccinating dogs against rabies prevents rabies in humans, and those costs should not be included in the total costs of rabies. Cost-of-illness studies generally include direct and indirect medical costs incurred once a disease or illness affects a population. They do not account for the costs associated with preventing the disease. Therefore, BCG vaccine costs are not included in the cost of TB, or COVID vaccine costs are not included in the cost of COVID, or seat belt costs are included in the cost of accidental injuries or deaths. The rabies cases occurred despite 3475 dogs being vaccinated. Dog vaccination costs are relevant when designing rabies elimination programs or prevention. Hence, it cannot be included in the costs of rabies. Though Shwiff et al (ref no 28) have included it in their paper , I strongly recommend not including it in this paper. If required, the dog vaccination costs can be presented separately.

**Results**

-Does the analysis presented match the analysis plan?

-Are the results clearly and completely presented?

-Are the figures (Tables, Images) of sufficient quality for clarity?

Reviewer #1: The analysis matches the plan and the results are clearly and completely presented. The authors have provided the raw data in an Excel spreadsheet as supplementary material. Tables and images are of sufficient quality for clarity.

Reviewer #2: Please include uncertainity intervals (see comments in attached file)

**Conclusions**

-Are the conclusions supported by the data presented?

-Are the limitations of analysis clearly described?

-Do the authors discuss how these data can be helpful to advance our understanding of the topic under study?

-Is public health relevance addressed?

Reviewer #1: The conclusions are supported by the data presented, and the limitations of the study are clearly described. Public health relevance is addressed.

Reviewer #2: Please see comments in attached file to improve this section.

**Editorial and Data Presentation Modifications?**

Reviewer #1: Line 63: Rabies is caused by any member of the Lyssavirus genus, not just 'a' member. This sentence can be rewritten as "Rabies is a neglected tropical disease caused by members of the Lyssavirus genus in the Rhabdoviridae family"

In general, clarity of the manuscript could be enhanced by additional editing from grammar and syntax.

Reviewer #2: Please see attached file.

**Summary and General Comments**

Reviewer #1: This study is a laudable effort to obtain data on the burden of canine rabies using a survey of households. Nationally reported data on human and animal rabies cases in LMICs is assumed to underestimate the true disease burden but by how much is not known. Studies such as these can help fill the gap and I would like to see more done. My comments below are mainly directed at improving rigor in methodology so that the study can be more readily replicated.

1. How were households within Kebeles selected? The manuscript states 'systematically' but this does not provide enough detail either for replication or to judge potential for selection bias. The method of household selection within Kebeles must be more fully described.

2. What proportion of respondents were excluded because of lack of familiarity with rabies? If substantial, this could underestimate the burden of rabies cases. The number and proportion of respondents excluded for this or other reasons should be reported in the results, perhaps in the form of a participant flow diagram if the number and reasons for exclusion warrant.

3. There seem to be two slightly different versions of the questionnaire: one included as Appendix 1 and another as S2 supplemental file. The Appendix 1 seems to map most directly onto the raw data in S1 and so I assume this is the correct version.

4. The authors seem to be using a de facto case definition of canine rabies based on typical symptoms as 1. A dog disease with aggressiveness and salivation, 2. A dog disease with paralysis and salivation, or 3. A dog disease with restlessness and salivation. If so, this could be summarised and stated in the methods: We defined a clinically suspected case as a dog with hypersalivation plus aggressiveness and/or restlessness and/or paralysis. The authors should note that this differs from the WHO definition of a suspect cases (being an animal that presents with any of the following clinical signs: Hypersalivation; Paralysis; Lethargy; Unprovoked or abnormal aggression (biting two or more people or animals, and/or inanimate objects); Abnormal vocalization; Diurnal activity of nocturnal species) but that their definition may provide a better working definition for household reporting (on the flip side, their definition would miss cases that do not show hypersalivation, or that show lethargy or abnormal vocalization in addition to hypersalivation without restlessness, aggression or paralysis). The authors must present their case definition clearly and discuss its pros and cons compared to the standard WHO case definition.

5. The authors do not define 'risky' vs. 'less risky' exposure. Here, they should follow the WHO classification of exposures as Category I (Touching or feeding animals, licks on intact skin (no exposure)), Category II (Nibbling of uncovered skin

Minor scratches or abrasions without bleeding (exposure)), or Category III (Single or multiple transdermal bites or scratches, contamination of mucous membrane or broken skin with saliva from animal licks, exposures due to direct contact with bats (severe exposure)). The authors category of 'less risky exposure' seems to include no contact with the dog - this could be a 'Category 0'.

6. The authors should report the annual incidence of 'risky exposures' to suspect rabid dogs per 100,000 people in the Results and compare it to similar estimates from other studies in the Discussion (ideally in the context of a standard definition of 'risky exposure' as WHO Category II/III or Category I/II/III, depending on what underlying criteria were used).

7. I'm confused about the animal exposures in the livestock economic analysis. In the Excel sheet of the raw data, all 85 exposed animals (86 is we count the one caprine) are classified as 'rabid' in the status variable (in the questionnaire, the options seems to be 'rabid' or 'suspect'). Does this mean they were laboratory confirmed? But if the 63 exposed bovines were culled after exposure, how could they have been confirmed? Did all the equines go on to develop clinical signs of rabies? Were they confirmed or suspect? It is very unlikely that all exposed animals would develop disease - in fact, the authors say in lines 299-300 that 'Most of the animals bitten by a suspect rabid dog did not develop rabies", yet in lines 304-305 they say "Thus, all exposed animals ended up with death as a result of the disease..." (referring to donkeys, which were not culled after exposure because of the lack of salvage value). The authors need to clarify this section.

Reviewer #2: (No Response)

PLOS authors have the option to publish the peer review history of their article (what does this mean?). If published, this will include your full peer review and any attached files.). If published, this will include your full peer review and any attached files.). If published, this will include your full peer review and any attached files.). If published, this will include your full peer review and any attached files.

...

Reviewer #1: No

Reviewer #2: No

**Figure resubmission:**

While revising your submission, we strongly recommend that you use PLOS’s NAAS tool (https://ngplosjournals.pagemajik.ai/artanalysis) to test your figure files. NAAS can convert your figure files to the TIFF file type and meet basic requirements (such as print size, resolution), or provide you with a report on issues that do not meet our requirements and that NAAS cannot fix.  After uploading your figures to PLOS’s NAAS tool - https://ngplosjournals.pagemajik.ai/artanalysis, NAAS will process the files provided and display the results in the "Uploaded Files" section of the page as the processing is complete. If the uploaded figures meet our requirements (or NAAS is able to fix the files to meet our requirements), the figure will be marked as "fixed" above. If NAAS is unable to fix the files, a red "failed" label will appear above. When NAAS has confirmed that the figure files meet our requirements, please download the file via the download option, and include these NAAS processed figure files when submitting your revised manuscript.
---

## [Decision Letter · Decision Letter 1]

26 Mar 2026

Dear Dr. Molla,

We are pleased to inform you that your manuscript 'Health and Economic Burden of Canine Rabies in Chilga District, Centeral Gondar Zone, Amhara Region, Ethiopia' has been provisionally accepted for publication in PLOS Neglected Tropical Diseases.

Best regards,

David Safronetz, Ph.D.

Section Editor

David Safronetz

Section Editor

Shaden Kamhawi

co-Editor-in-Chief

Paul Brindley

co-Editor-in-Chief

Reviewer's Responses to Questions

**Key Review Criteria Required for Acceptance?**

**Methods**

-Are the objectives of the study clearly articulated with a clear testable hypothesis stated?

-Is the study design appropriate to address the stated objectives?

-Is the population clearly described and appropriate for the hypothesis being tested?

-Is the sample size sufficient to ensure adequate power to address the hypothesis being tested?

-Were correct statistical analysis used to support conclusions?

-Are there concerns about ethical or regulatory requirements being met?

Reviewer #1: (No Response)

Reviewer #2: None

**Results**

-Does the analysis presented match the analysis plan?

-Are the results clearly and completely presented?

-Are the figures (Tables, Images) of sufficient quality for clarity?

Reviewer #1: (No Response)

Reviewer #2: The cost of rabies should be expressed per year and not per two years throughout the manuscript. It is easier to compare annual costs which is what conventionally used as default in literature.

**Conclusions**

-Are the conclusions supported by the data presented?

-Are the limitations of analysis clearly described?

-Do the authors discuss how these data can be helpful to advance our understanding of the topic under study?

-Is public health relevance addressed?

Reviewer #1: (No Response)

Reviewer #2: None

**Editorial and Data Presentation Modifications?**

Reviewer #1: (No Response)

Reviewer #2: The authors have used informal words and phrases in some places . this needs to be corrected and made more professional. For eg

Ethiopia is one of the worst rabies affected can be Ethoipis is a high burden rabies country in world or Ethopia is among the top 10 countries affected etc.

Price 2 for equines is obviously zero as rabid equines have no salvage value to Price 2 for equines was zero since it had no salvage value.

Sentences should appear more proefessional.

**Summary and General Comments**

Reviewer #1: The authors have adequately addressed the comments I made in my review.

Reviewer #2: The revised version has come out really well. The authors have addresed most of the points I have raised.

PLOS authors have the option to publish the peer review history of their article (what does this mean?). If published, this will include your full peer review and any attached files.). If published, this will include your full peer review and any attached files.). If published, this will include your full peer review and any attached files.). If published, this will include your full peer review and any attached files.

...

Reviewer #1: No

Reviewer #2: No

---

## [Editor Report · Acceptance letter]

Dear Dr. Molla,

We are delighted to inform you that your manuscript, "Health and Economic Burden of Canine Rabies in Chilga District, Centeral Gondar Zone, Amhara Region, Ethiopia," has been formally accepted for publication in PLOS Neglected Tropical Diseases.

Best regards,

Shaden Kamhawi

co-Editor-in-Chief

Paul Brindley

co-Editor-in-Chief
